# Influence of Different Glycoproteins and of the Virion Core on SERINC5 Antiviral Activity

**DOI:** 10.3390/v13071279

**Published:** 2021-06-30

**Authors:** William E. Diehl, Mehmet H. Guney, Teresa Vanzo, Pyae P. Kyawe, Judith M. White, Massimo Pizzato, Jeremy Luban

**Affiliations:** 1Program in Molecular Medicine, University of Massachusetts Medical School, 373 Plantation Street, Worcester, MA 01605, USA; toktora@gmail.com (W.E.D.); mehmet.guney@umassmed.edu (M.H.G.); pyae.kyawe@umassmed.edu (P.P.K.); 2Department of Cellular, Computational and Integrative Biology, University of Trento, 38123 Trento, Italy; teresa.vanzo@unitn.it (T.V.); massimo.pizzato@unitn.it (M.P.); 3Department of Cell Biology, University of Virginia, Charlottesville, VA 22908, USA; jw7g@virginia.edu; 4Department of Biochemistry and Molecular Pharmacology, University of Massachusetts Medical School, Worcester, MA 01605, USA

**Keywords:** retroviruses, glycoproteins, restriction factor, SERINC5, gag, virion, pseudotype

## Abstract

Host plasma membrane protein SERINC5 is incorporated into budding retrovirus particles where it blocks subsequent entry into susceptible target cells. Three structurally unrelated proteins encoded by diverse retroviruses, human immunodeficiency virus type 1 (HIV-1) Nef, equine infectious anemia virus (EIAV) S2, and ecotropic murine leukemia virus (MLV) GlycoGag, disrupt SERINC5 antiviral activity by redirecting SERINC5 from the site of virion assembly on the plasma membrane to an internal RAB7+ endosomal compartment. Pseudotyping retroviruses with particular glycoproteins, e.g., vesicular stomatitis virus glycoprotein (VSV G), renders the infectivity of particles resistant to inhibition by virion-associated SERINC5. To better understand viral determinants for SERINC5-sensitivity, the effect of SERINC5 was assessed using HIV-1, MLV, and Mason-Pfizer monkey virus (M-PMV) virion cores, pseudotyped with glycoproteins from Arenavirus, Coronavirus, Filovirus, Rhabdovirus, Paramyxovirus, and Orthomyxovirus genera. SERINC5 restricted virions pseudotyped with glycoproteins from several retroviruses, an orthomyxovirus, a rhabdovirus, a paramyxovirus, and an arenavirus. Infectivity of particles pseudotyped with HIV-1, amphotropic-MLV (A-MLV), or influenza A virus (IAV) glycoproteins, was decreased by SERINC5, whether the core was provided by HIV-1, MLV, or M-PMV. In contrast, particles pseudotyped with glycoproteins from M-PMV, parainfluenza virus 5 (PIV5), or rabies virus (RABV) were sensitive to SERINC5, but only with particular retroviral cores. Resistance to SERINC5 did not correlate with reduced SERINC5 incorporation into particles, route of viral entry, or absolute infectivity of the pseudotyped virions. These findings indicate that some non-retroviruses may be sensitive to SERINC5 and that, in addition to the viral glycoprotein, the retroviral core influences sensitivity to SERINC5.

## 1. Introduction

HIV-1 Nef is important for maximal virus replication in vivo and for progression to AIDS [1,2,3]. Nef is a multifunctional accessory protein that downregulates CD4, MHC Class I, and other molecules from the cell surface [4,5,6,7,8,9]. Nef also enhances HIV-1 infectivity in single-round infection experiments [10,11,12,13,14,15,16,17] by overcoming the antiviral effects of SERINC5 and SERINC3 [18,19], though, of the two, SERINC5 is the more potent restriction factor. SERINC5 is incorporated into budding virions where it inhibits subsequent fusion of the virion membrane with target cell membranes. Nef counteracts SERINC5 by removing it from the cell surface so that it is not incorporated into nascent virions [18,19,20,21].

HIV-1 is not the only virus inhibited by SERINC5. Simian immunodeficiency viruses (SIV) lacking nef are also inhibited by SERINC5 and SIV nef counteracts this inhibition [18] with a potency that is proportional to the prevalence of SIV in feral primate populations [22]. Two examples of convergent evolution of anti-SERINC function by virally encoded proteins are found outside of primate immunodeficiency viruses. MLV GlycoGag and EIAV S2 are viral antagonists of SERINC5 activity, and neither share sequence or structural homology with Nef, nor to each other [18,23,24,25].

The mechanism by which virion-associated SERINC5 inhibits HIV-1 entry is unknown. The block is manifest after virion attachment to target cells, apparently at the stage of fusion pore expansion; virion contents mix with target cell cytoplasm but virion core transfer to the cytoplasm is inhibited [18,20,26]. Virions that are otherwise isogenic exhibit a range of dependency on Nef and of sensitivity to SERINC5 when pseudotyped with HIV-1 Env glycoproteins from different HIV-1 isolates [27,28]. SERINC5 increases HIV-1 sensitivity to antibodies and peptides targeting the membrane-proximal external region (MPER) of HIV-1 TM/gp41, suggesting that it somehow alters the conformation of the HIV-1 glycoprotein near the virion membrane [20,27]. Importantly, HIV-1 particles pseudotyped with VSV G or Ebola virus (EBOV) glycoprotein are resistant to SERINC5 antiviral activity [18,19,25]. These initial observations suggest a correlation between the location of viral fusion and sensitivity to SERINC5 activity, with glycoproteins that mediate fusion at the cell surface (Env from HIV-1 and A-MLV) being sensitive and those that mediate fusion in endo-lysosomal compartments (VSV and EBOV) being resistant [18,25], though one study is not consistent with this idea [29]. Taken together these results indicate that the virion glycoprotein is a viral determinant of sensitivity to SERINC5.

SERINC5 is a multipass transmembrane protein that localizes almost exclusively to the plasma membrane [18,19]. As such, in the absence of countermeasures, all enveloped viruses that bud through the plasma membrane would be expected to encounter SERINC5 during viral egress, and to potentially be subject to its antiviral effects. We sought to address the breadth of SERINC5 antiviral activity and assess whether the route of entry impacts the sensitivity of viral glycoproteins to the antiviral effects of SERINC5. To do so, we investigated whether the co-expression of SERINC5 during viral production could inhibit a variety of glycoprotein pseudotypes of HIV, MLV, or M-PMV cores. Using this system, we tested the sensitivity of a number of retroviral Env glycoproteins as well as representative glycoproteins from the Arenavirus, Coronavirus, Filovirus, Rhabdovirus, Paramyxovirus, and Orthomyxovirus families. Consistent with previous findings, we observed that the glycoprotein is a major determinant of SERINC5 sensitivity. While many glycoproteins were insensitive to the antiviral effects of SERINC5 under all conditions tested here, pseudotypes with the glycoproteins from HIV-1NL4-3, A-MLV, and IAV were inhibited by SERINC5 in the context of all virion cores tested. No correlation was observed between SERINC5 sensitivity and the route of viral entry mediated by the viral glycoprotein. Unexpectedly, we also observed that sensitivity to SERINC5 antiviral activity for M-PMV, PIV5, and RABV glycoproteins depended on the retroviral core onto which they were pseudotyped. Our findings reveal that the interplay between virion core and glycoprotein is a determinant of virion sensitivity to SERINC5 antiviral activity.

## 2. Materials and Methods

### 2.1. Plasmid DNA

Plasmids used in this study are described in Appendix A, including Addgene or NIH AIDS Reagent Program code numbers (where applicable), where full plasmid sequences can be obtained. A pcDNA3.1 based vector bearing codon-optimized pNL4-3 env with a cytoplasmic tail truncation after residue 710 (HXB2 residue 712), similar to that previously described [30], was generated using standard cloning techniques and is available from Addgene.

### 2.2. Cell Culture

HEK293 cells were obtained from the ATCC. The HIV indicator cell line TZM-bl was obtained from the AIDS Research and Reference Reagent Program (Cat#8129, Division of AIDS, NIAID, NIH) and were deposited by Drs. John C. Kappes and Xiaoyun Wu [31]. Both cell lines were maintained in DMEM supplemented with 10% FBS and 10 mM HEPES.

### 2.3. Virus Production, and Transductions

All viral stocks were generated by Mirus TransIT-LT1 (Mirus Bio, Madison, WI, USA) mediated transfection of HEK293 cells. Twelve-well plates were seeded with 3 × 10^5^ cells per well 24 h prior to transfection. Then, 3.375 μL LT1 reagent was used to transfect plasmids as follows: For the production of pseudotyped HIV-1 virions 625 ng pNL-EGFP/CMV-WPRE∆U3 [32] and 465 ng pCD/NL-BH*∆∆∆ [33] were co-transfected with 155 ng glycoprotein expression vector. For the production of pseudotyped MLV virions 625 ng pLXIN-GFP [34] and 465 ng pCS2+mGP [35] were co-transfected with 155 ng glycoprotein expression vector. For the production of pseudotyped M-PMV virions, 1090 ng pSARM-EGFP [32] was co-transfected with 155 ng glycoprotein expression vector. In all cases, either 100 ng pcDNA-SERINC5 [18] or 110 ng empty pcDNA3.1 (Thermo Fisher Scientific, Waltham, MA, USA) vector was included in these transfections. After 16 h, transfection medium was replaced with fresh DMEM and virus containing supernatant was harvested 48 h after media change. This supernatant was spun for 10 min at 2500× *g* to remove cellular debris and stored at 4 °C until used for transduction.

HEK293 or TZMbl cells were seeded at 1 × 10^5^ or 5 × 10^4^, respectively, in 12-well plates 24 h prior to transduction. For experiments involving ecotropic MLV or avian leukosis virus A, HEK293 cells were transfected in 6-well plates with 2.5 μg of pBABE-puro-mCAT or pCMMP-TVA800 using TransIT-LT1 and the subsequent day these transfected cells were split and plated for transductions. For transductions, culture supernatant was replaced with three dilutions of virus containing supernatant and incubated at 37 °C. Virus containing medium was replaced at 16 h and cells were incubated for an additional 48 h, following which they were trypsinized and assessed for GFP expression via fluorescent activated cell sorting using the Accuri C6 (BD Biosciences, San Jose, CA, USA). Analysis was performed using FlowJo Macintosh v10.1 (FlowJo, LLC, Ashland, OR, USA).

### 2.4. Virion Purification and Western Blotting

Viral pseudotypes were produced as above, except transfections were performed in 6-well plates so the number of cells plated and DNA introduced were doubled. The resulting virus-containing supernatant was overlayed on 20% sucrose in TNE buffer (50 mM TRIS, 100 mM NaCl, 0.1 mM EDTA, pH7.4) and viruses were pelleted via ultracentrifugation for 2 h at 125,000× *g* at 4 °C using an SW55-Ti rotor (Beckman Coulter, Indianapolis, IN, USA). Following centrifugation, tubes were washed with 1 mL of ice cold PBS and viral pellets were directly lysed in 50 μL 2× Laemmli buffer containing 50 mM TCEP [Tris(2-carboxyethyl)phosphine] incubated at room temp for 5 min. Cell lysates were prepared in parallel by washing transfected HEK293s once with 1 mL ice cold PBS, detaching from the plate by scraping, pelleting, and subsequently lysing for 20 min on ice in 150 μL SERINC lysis buffer (10 mM HEPES, pH 7.5, 100 mM NaCl, 1 mM TCEP [Tris(2-carboxyethyl)phosphine], 1% DDM [n-Dodecyl-β-d-maltoside]) containing cOmplete mini protease inhibitor (Sigma-Aldrich, St. Louis, MO, USA). Lysates were clarified by centrifugation for 5 min at 10,000× *g* and 4 °C, following which supernatants were transferred to a new centrifuge tube and protein content was quantified via Reducing Agent Compatible BCA Assay (Thermo Scientific, Waltham, MA, USA) Volumes of lysate corresponding to equal protein content were combined 1:1 with 2× Laemmli buffer containing 50 mM TCEP and incubated at room temp for 5 min.

One half of the denatured viral pellet and approximately 8 μg protein from cellular lysates were run on 4–15% gradient acrylamide gels, and transferred to nitrocellulose membranes. SERINC5 levels were assessed via C-terminal HA tag using the mouse monoclonal HA.11 (Biolegend, San Diego, CA, USA) at 1 μg/mL in Odyssey blocking buffer (LI-COR Biotechnology, Lincoln, NE, USA). HIV-1 p24 was detected using human monoclonal antibody 241-D [36] at a concentration of 1 μg/mL in Odyssey blocking buffer. MLV p30 was detected with rat monoclonal antibody R187 [37] from unpurified culture medium following five days of culturing the R187 hybridoma (ATCC, Manassas, VA, USA). This medium was diluted 1:200 in Odyssey blocking buffer. Cellular actin was detected using mouse anti-actin monoclonal ACTN05 (C4) (Abcam, Cambridge, MA, USA) at a concentration of 0.5 μg/mL in Odyssey blocking buffer. All blots were developed using 1:10,000 dilutions of 680RD or 800CW fluorescently tagged secondary antibodies (LI-COR Biotechnology, Lincoln, NE, USA) in Odyssey blocking buffer. Imaging of blots was performed using an Odyssey CLx system (LI-Cor Biotechnology) at a resolution of 84 μm using the ‘high quality’ setting. Quantitation of bands was done using the box tool in the Odyssey software package with adjacent pixels to the box serving as reference background levels for background subtraction.

### 2.5. Statistics

All statistical analysis was performed with Prism v8.3.0 (GraphPad Software, LLC, San Diego, CA, USA) and all tests were considered statistically significant when *p* < 0.05.

For all calculations of statistical significance the ‘Fold Reduction in Titer with SERINC5’ values were transformed to log10. This transformation was considered an appropriate approach [38] since the variances of the data were unequal, with the variance being proportional to the magnitude of the observed restriction—owing to the fact that the data were a ratio of infectivities.

To determine the statistical significance of SERINC5 sensitivity of pseudotypes of a given viral core, one-way ANOVA was performed with Dunnett’s multiple comparisons post-test for each pseudotype against an idealized no restriction control. This control set consisted of 4 randomly generated data points with an average (log10 value) of 0. Further, the standard deviation of this control set was designed to be equal to the average standard deviation of the fold reduction in infectivity with SERINC5 values from the 60 glycoprotein/core combinations tested (0.21328). We performed additional statistical analyses on those glycoproteins identified as being differentially sensitive to SERINC5 antiviral activity when pseudotyped on the different viral cores. For each glycoprotein, Brown-Forsythe and Welsh ANOVA tests with Dunnett’s T3 multiple comparisons post-test were used to compare each viral core against both others.

The Grubbs test was used to identify outlier data points (*p* < 0.05). Single data points that were identified as outliers and excluded from further analysis were in the following datasets: SARS CoV and EcoMLV pseudotypes of HIV-1 cores, SARS CoV and HTLV-1 pseudotypes of MLV cores, and LCMV, RD114, and EcoMLV pseudotypes of M-PMV cores.

Linear regression analysis was used to measure correlation between SERINC5 incorporation and the average SERINC5-restriction.

## 3. Results

To determine which viral glycoproteins are sensitive to the antiviral activity of SERINC5, pseudotyped GFP-expressing HIV-1 vectors were produced in the presence or absence of SERINC5, and infectivity was assessed. Since antiviral activity correlates with the level of SERINC5 expression [18], to minimize false negatives, SERINC5 was expressed from the relatively strong CMV promoter. The panel of glycoproteins examined for SERINC5 sensitivity included a diverse selection of retroviral Env glycoproteins, including those encoded by HIV-1, avian leukosis virus A (ALV-A), human endogenous retrovirus K (HERV-K), feline endogenous retrovirus RD114, M-PMV, ecotropic MLV (EcoMLV), A-MLV, and human T-cell lymphoma virus-1 (HTLV-1). We also tested the glycoproteins from an assortment of RNA viruses including IAV (H7/N1), PIV5, measles virus, RABV, lymphocytic choriomeningitis virus (LCMV), Marburg virus (MARV), EBOV Zaire [Mayinga], severe acute respiratory virus coronavirus (SARS CoV), and VSV.

SERINC5 caused a greater than 100-fold reduction in viral infectivity for HIV-1 and A-MLV Env pseudotypes, while no significant reduction was observed for EBOV or VSV glycoprotein pseudotypes (Figure 1a and Table 1). Nearly identical results were observed for HIV-1 Env when the cytoplasmic tail was truncated after residue 710 (Figure 2), a mutation that disrupts interactions between the viral TM protein and the virion core [39]. Significant reduction in infectivity was also observed for pseudotypes with the glycoproteins encoded by M-PMV, A-MLV, PIV5, RD114, IAV H7/N1, RABV, and LCMV. These observations indicate that SERINC5 restriction is not dictated by where in the cell the viral glycoprotein mediates fusion, since fusion mediated by IAV [40], RABV [41], and LCMV [42] occurs in a pH-dependent fashion in the endo-lysosomal compartment, while fusion mediated by HIV-1 [43,44], A-MLV, M-PMV [45], or PIV5 [46] occurs in a pH-independent manner at the cell surface (Figure 3).

Next we tested a panel of filoviral glycoproteins for sensitivity to SERINC5 restriction. All of these glycoproteins require proteolytic processing [47,48] following internalization into the target cell and utilize the lysosomal protein NPC1 to initiate viral fusion [49,50]. In addition to the EBOV and MARV glycoproteins tested in Figure 1a, this panel included glycoproteins from Bundibugyo (BDBV), Lloviu (LLOV), Reston (RESTV), Sudan (SUDV), Taï Forest (TAFV), and the 2014 Makona glycoprotein variant (A82) that initiated the 2013-2016 outbreak, along with an infectivity-enhancing variant (GP-A82V) that arose during the outbreak [51,52]. The magnitude SERINC5 inhibition did not achieve statistical significance for any of the filoviral glycoproteins (Figure 1b). Among them, though, there appeared to be modest differences in sensitivity. The RESTV and TAFV glycoproteins appeared to be slightly more sensitive (4.3- and 2.9-fold, respectively) to SERINC5 inhibition than were the glycoproteins of either Mayinga or Makona EBOV (1.65- and 1.2-fold, respectively).

HIV-1 Nef, MLV GlygoGag, and EIAV S2 counteract SERINC5 antiviral activity by removing SERINC5 protein from the cell surface and relocalizing it to an endosomal compartment [18,19,23]. Given that HIV-2 Env inhibits the human antiviral protein BST2 by downregulating it from the cell surface [53], a particular viral glycoprotein might confer resistance to SERINC5 antiviral activity by internalizing it. That being said, there was no obvious decrease in SERINC5 incorporation into virions in the presence of any of the viral glycoproteins (Figure 4a,b). In fact, virions bearing glycoproteins from SARS CoV or VSV, two glycoproteins that are resistant to SERINC5, incorporated at least twice as much SERINC5 as did particles pseudotyped with HIV-1 or A-MLV, two glycoproteins that are sensitive to SERINC5. When SERINC5 incorporation into particles with all seventeen glycoproteins was considered, no correlation between SERINC5 exclusion from virions and resistance to its antiviral effects was evident (Figure 4c).

In retroviruses, glycoprotein incorporation into virions is dependent on the identity of the virion core [54] and dictated in part by interaction between the glycoprotein TM subunit and gag-encoded MA [55,56]. We therefore tested the effect of SERINC5 when the same glycoproteins tested on HIV-1 cores (Figure 1a) were pseudotyped on MLV virion cores. Several glycoproteins that were sensitive to SERINC5 restriction on HIV-1 cores were also restricted when pseudotyped on MLV cores (the glycoproteins of HIV-1, A-MLV, IAV, and RABV; Figure 5a and Table 1). In contrast to what was observed with HIV-1 cores, pseudotypes with LCMV glycoprotein were not inhibited by SERINC5 to a significant extent when incorporated on MLV cores (Figure 5a and Table 1). Though RD114 Env pseudotyped on MLV cores was restricted ~7-fold, the magnitude of the inhibitory effect on MLV cores did not reach statistical significance (Figure 5a and Table 1). Although M-PMV Env was not sensitive to SERINC when pseudotyped on HIV-1, it was sensitive when pseudotyped onto cores from MLV (Figure 5a and Table 1).

Analysis of the panel of glycoproteins was then extended to M-PMV cores (Figure 5b and Table 1), on which infectivity with M-PMV Env was reduced significantly in the presence of SERINC5. Pseudotypes of M-PMV cores with HIV-1, A-MLV, and IAV glycoproteins were sensitive to SERINC5 restriction, similar to these pseudotypes produced on HIV-1 and MLV cores. RD114 Env pseudotypes of M-PMV cores were sensitive to SERINC5 restriction as was the case with HIV-1 cores. In contrast to results with HIV-1 cores, LCMV pseudotypes on M-PMV cores were insensitive to SERINC5 antiviral activity. RABV glycoprotein pseudotypes on M-PMV cores were not significantly affected by the antiviral effects of SERINC5, which differs from what was observed on HIV-1 and MLV cores. Finally, PIV5 was uniquely sensitive to the antiviral effects of SERINC5 when it was pseudotyped on M-PMV cores.

Of the seventeen glycoproteins tested here, RD114, M-PMV, RABV, PIV5, and LCMV displayed sensitivity to SERINC5 antiviral activity in a manner that depended on the viral core. To determine if these core-specific differences were significant, the SERINC5 inhibitory effect for pseudotypes on each core were assessed against every other core using analysis of variance (ANOVA). By these parameters, M-PMV pseudotypes on HIV-1 cores were less sensitive to SERINC5 than were pseudotypes based on either MLV or M-PMV cores (Table 2). Additionally, the analysis determined that PIV5 and RABV glycoprotein pseudotypes built on M-PMV cores were statistically more sensitive to SERINC5 antiviral effects than when these same glycoproteins were displayed on HIV-1 cores (Table 2). Finally, the SERINC5 sensitivity of RD114 and LCMV glycoprotein pseudotypes did not differ significantly on different cores (Table 2).

## 4. Discussion

Initial reports demonstrated that the viral glycoprotein is a determinant of sensitivity to SERINC5 antiviral activity [18,19,27,28] and suggested that viral glycoproteins mediating fusion via a pH-dependent, endocytic entry pathway are resistant to SERINC5 antiviral activity [18,25]. Here, we examined the SERINC5 susceptibility of glycoproteins from diverse families of enveloped viruses. Virus production from HEK293 cells in the presence of ectopically expressed SERINC5 showed that SERINC5 inhibits virions pseudotyped with glycoproteins from several retroviruses (HIV-1, A-MLV, RD114, and M-PMV), an orthomyxovirus (IAV), a rhabdovirus (RABV), a paramyxovirus (PIV5), and an arenavirus (LCMV). Nef exerts its effects within virion producer cells [17,57], so infectivity on a range of target cells was not examined here, though relative SERINC5 restriction activity was nearly identical when HEK-293 or NIH-3T3 were used as target cells (W.E. Diehl and J. Luban, unpublished data). To our knowledge, these experiments are the first time that SERINC5 restriction of non-retroviral glycoproteins has been detected. Further studies will be needed to determine whether the glycoproteins identified here as being restricted by SERINC5 in the context of pseudotypes—including IAV, RABV, PIV5, and LCMV—are inhibited by endogenous SERINC5 in the context of autologous virions.

In contrast to expectations based on the SERINC5 resistance of pseudotypes with the VSV and EBOV glycoproteins [18,19,58,59], the results reported here with IAV, RABV, and LCMV glycoproteins demonstrate that a low pH-dependent endocytic entry pathway does not preclude sensitivity to the antiviral effects of SERINC5. Strain-specific differences in sensitivity to SERINC5 restriction exist for HIV-1 [18,27,28], and might exist for other viruses, but none were detected for EBOV (Figure 1b) and the broad survey of virus glycoproteins that was conducted here generally did not test multiple strains for each virus. Additionally, sensitivity to SERINC5 might be expected to correlate inversely with levels of glycoprotein incorporation on virions. This however seems unlikely since the number of Env trimers incorporated from SERINC5-sensitive HXB2, NL4-3, and SF162 strains of HIV-1 is nearly identical to the number of trimers incorporated from SERINC5-resistant JRFL, ADA, and YU2 strains of HIV-1 [18,27,28,60,61]. One might expect that the magnitude SERINC5 sensitivity of the heterologous pseudoparticles used here would correlate with absolute infectivity (Table 1). However, this was not the case since linear regression slope values comparing SERINC5 sensitivity with absolute infectivity for the different glycoproteins on each of the virion cores were small values approaching zero.

While retroviral Env glycoproteins from HIV-1, MLV, and RD114 have all previously been found to be inhibited by SERINC5 [18,19], we now report that M-PMV glycoprotein is SERINC5-sensitive as well. Given that other retroviruses encode anti-SERINC5 proteins, the ~100-fold reduction in infectivity of autologously pseudotyped M-PMV cores produced in the presence of SERINC5 is not what might have been expected for what was an essentially complete provirus. If the simian virus M-PMV does encode a SERINC5 antagonist, perhaps ectopic production of SERINC5 saturated its activity, or this putative M-PMV factor lacks the ability to inhibit SERINC5 in non-native human cells.

Susceptibility of particular glycoproteins to the antiviral effects of SERINC5 depended on the virion core. Comparison of SERINC5-mediated restriction with an idealized control (see statistics Section 2 for details) identified M-PMV, PIV5, RD114, RABV, and LCMV glycoproteins as differentially sensitive to SERINC5 inhibition (Figure 1a, Figure 3a,b, and Table 1). However, when the magnitude of SERINC5 restriction was assessed for a given glycoprotein across the three virions cores, differences were statistically significant for only M-PMV, PIV5, and RABV glycoproteins (Table 2). Specifically, M-PMV glycoprotein was sensitive to SERINC5 restriction when on MLV or M-PMV cores, but resistant when on HIV-1 cores (Figure 1a, Figure 3a,b, and Table 1 and Table 2). In contrast, RABV glycoprotein was sensitive to SERINC5 restriction when on HIV-1 or MLV cores, but not on M-PMV cores (Figure 1a, Figure 3a,b, and Table 1), but only RABV glycoprotein pseudotypes on HIV-1 and M-PMV cores differed significantly (Table 2). Similarly, PIV5 glycoprotein pseudotypes on M-PMV cores, but not on HIV-1 or MLV cores, were sensitive to SERINC5 antiviral activity (Figure 1a, Figure 3a,b, and Table 1), though only the pseudotypes on HIV-1 and M-PMV cores differed significantly in their sensitivities to SERINC5 (Table 2). In contrast, while LCMV glycoprotein pseudotypes on HIV-1 cores were sensitive to SERINC5 restriction, the difference in magnitude of the restriction was not statistically significant when compared to pseudotypes on MLV or M-PMV. Finally, HIV-1 and M-PMV cores pseudotyped with RD114 Env were sensitive to SERINC5, while pseudotypes of MLV cores were not (Figure 1a, Figure 3a,b, and Table 1). However, no statistically significant differences in SERINC5 restriction were identified between these viral cores, a point that is consistent with a previous report demonstrating a magnitude of inhibition for RD114 Env-pseudotyped MLV cores by SERINC5 that was similar to what was reported here [25].

While the mechanism underlying the core-dependent antiviral activity of SERINC5 remains to be determined, several possible explanations can be envisioned. SERINC5 alters the sensitivity of HIV-1 virions to neutralization by monoclonal antibodies targeting the HIV-1 TM MPER [20,27]. Given that MA, the membrane proximal domain of the Gag polyprotein, contacts the retroviral TM [55,56], incorporation of SERINC5 into the virion membrane has the potential to influence interactions between MA and TM in the HIV-1 virion that are essential for infectivity. Similarly, SERINC5 might influence retroviral core interactions made by the heterologous glycoproteins tested here, for which SERINC5 restriction activity was core-dependent, i.e., the M-PMV, PIV5, and RABV glycoproteins (Table 2).

Alternatively, though the lipid composition of HIV-1 virions is not detectably altered by SERINC5 [62], there might be conditions in which SERINC5 antiviral activity is influenced by virion lipid composition. Retroviruses assemble at different subcellular locations and incorporate unique assemblages of lipids during budding. For instance, HIV-1 and MLV virions, both of which assemble and bud from cholesterol-rich microdomains of the plasma membrane, differ in the proportions of glycerophosphatidylinositol bisphosphate (PIP2), glycerophosphatidylethanolamine (PE), plasmalogen-glycerophosphatidylethanolamine (pl-PE), and glucosylceramide (Glu-Cer) that they contain [63]. The lipid composition of M-PMV virions is less well studied, but they are likely to have a different lipid makeup than HIV-1 or MLV. This is because M-PMV does not assemble at the plasma membrane like HIV-1 and MLV. Rather, it assembles immature particles in a perinuclear region of the cytosol and these preformed proteinaceous shells are then transported to the plasma membrane where the pre-assembled spherical protein shell is enveloped and released [64,65]. Additionally, the lipid composition of non-retroviruses such as IAV is critical for virion infectivity and may be a determinant of pathogenicity [66]. The differences in SERINC5 sensitivity for glycoprotein pseudotypes on particular cores (Table 2) might therefore be explained by differences in lipid content.

## Figures and Tables

**Figure 1 viruses-13-01279-f001:**
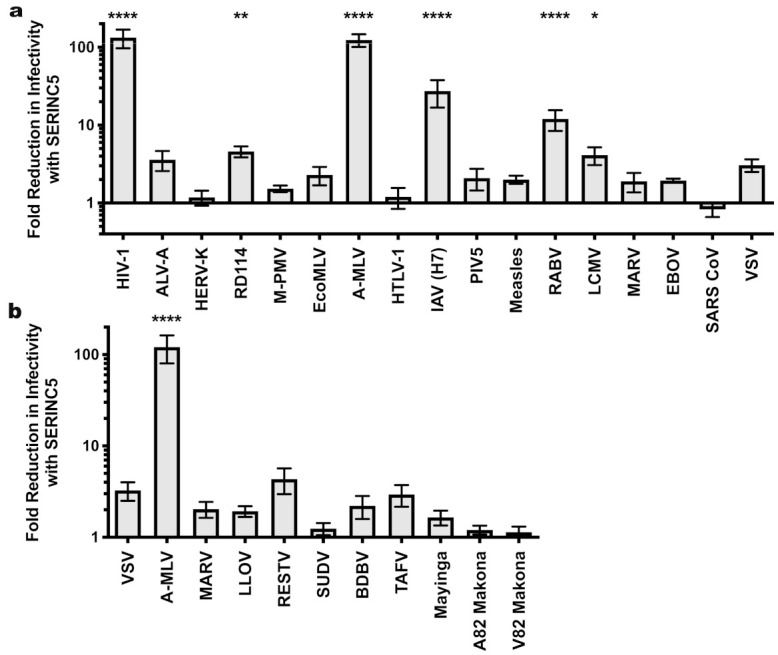
Sensitivity of HIV-1 pseudotypes to SERINC5 antiviral activity. (**a**) Effect of SERINC5 on transduction efficiency by HIV-1 cores pseudotyped with a diverse panel of viral glycoproteins. Transductions used HEK293 cells as target cells except that HIV-1 pseudotypes used TZMbl cells. HEK293 cell targets were transfected with the cognate viral receptor prior to transduction with ecotropic MLV or avian leukosis virus A pseudotypes. (**b**) Sensitivity of filoviral glycoprotein pseudoviruses to SERINC5. Plotted is the difference in infectivity between viruses produced in the absence versus the presence of SERINC5. Each condition shows results of vector production from at least three independent transfections. Statistical significance of observed SERINC5 effects was determined via one-way ANOVA with Dunnett’s multiple comparisons post-test as indicated in the Materials and Methods. *, *p* < 0.05; **, *p* < 0.01; ****, *p* < 0.0001. HIV-1: human immunodeficiency virus-1; ALV-A: avian leukosis virus A; HERV-K: human endogenous retrovirus K; RD114: feline endogenous retrovirus RD114; M-PMV: Mason-Pfizer monkey virus; EcoMLV: ecotropic MLV; AMLV: amphotropic MLV; HTLV-1: human T cell lymphotropic virus type 1; IAV: influenza A virus; PIV5: parainfluenza virus 5; RABV: rabies virus; LCMV: lymphocytic choriomeningitis virus; MARV: Marburg virus; EBOV: Mayinga isolate of Zaire Ebolavirus; SARS CoV: severe acute respiratory syndrome coronavirus; VSV: vesicular stomatitis virus; LLOV: Lloviu virus; RESTV: Reston virus; SUDV: Sudan virus; BDBV: Bundibugyo virus; TAFV: Taï Forest virus; Mayinga: Mayinga isolate of Ebola virus; Makona: Makona isolate of Ebola virus.

**Figure 2 viruses-13-01279-f002:**
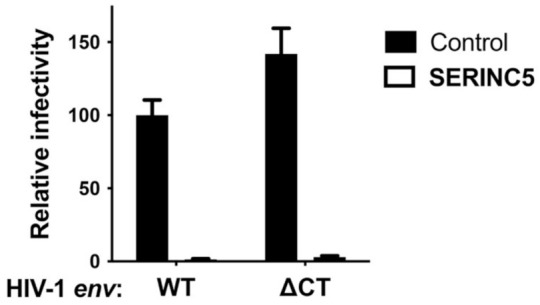
HIV-1 Env with a deletion in the cytoplasmic tail is sensitive to SERINC5 restriction activity. HIV-1 pseudotypes bearing HIV-1 Env, either WT or with deletion of amino acids after 710, were produced by co-transfection of HEK293 cells with a SERINC5 expression plasmid or control. Target cells were TZMbl cells. Each condition shows results of vector production from at least three independent transfections. Statistical significance of the observed SERINC5 effects was determined as in Figure 1, and in each case *p* < 0.0001.

**Figure 3 viruses-13-01279-f003:**
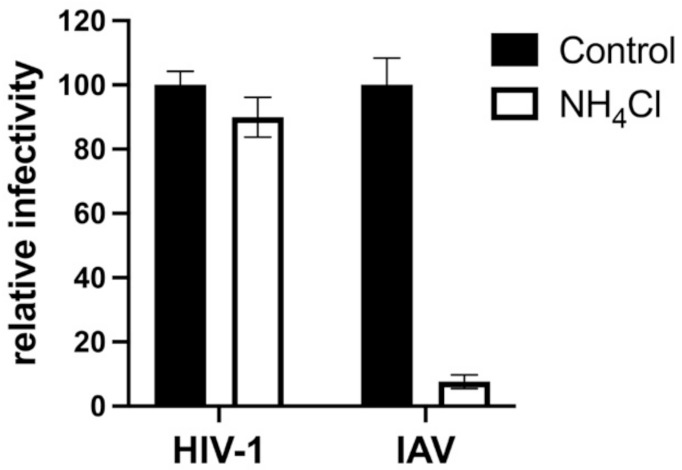
Transduction of TZMbl cells is pH-dependent when vectors are pseudotyped with glycoprotein encoded by IAV but not by HIV-1. HIV-1 virions pseudotyped with glycoproteins encoded by either HIV-1 or IAV H7/N1 were produced by transfection of HEK293 cells. TZMbl cells were then challenged with the pseudotyped vectors, either in the presence of NH4Cl or without. Each condition shows results of vector production from at least three independent transfections. Statistical significance of the observed effects of NH4Cl on IAV was *p* < 0.0001, determined as in Figure 1.

**Figure 4 viruses-13-01279-f004:**
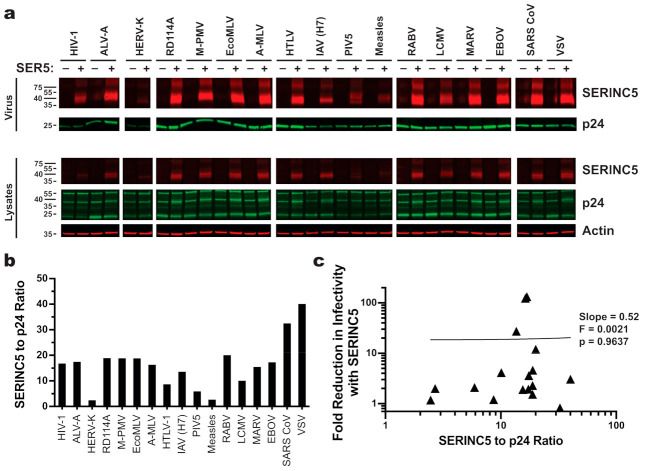
SERINC5 incorporation efficiency does not correlate with its antiviral activity. (**a**) [Top] Western blots of enriched HIV-1 pseudovirions produced in the presence or absence of C-terminally HA-tagged SERINC5. Blots were probed with mouse monoclonal anti-HA and human anti-p24 monoclonal 241-D. [Bottom] Western blots of lysates from the HEK293 cells that were used to produce the pseudovirions in the top panel. Blots were probed with mouse anti-actin in addition to anti-HA and anti-p24. (**b**) Quantitation of blots shown in panel A. The signal for SERINC5 incorporated into cell-free HIV-1 pseudovirions was normalized to p24. (**c**) Linear regression analysis of normalized amounts of SERINC5 incorporated into pseudovirions (**b**) versus the magnitude SERINC5 inhibition (**a**).

**Figure 5 viruses-13-01279-f005:**
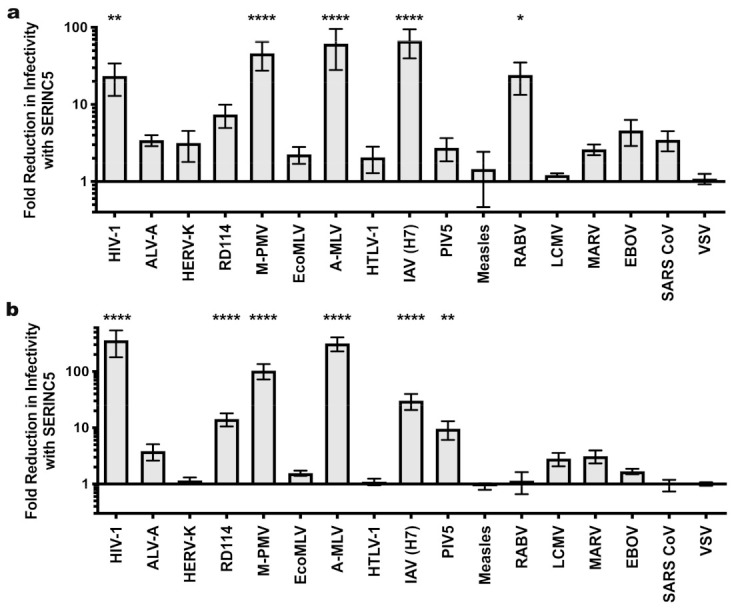
Sensitivity of MLV and M-PMV pseudotypes to SERINC5 antiviral activity. The effect of SERINC5 on the infectivity of glycoprotein pseudotypes using MLV (**a**) or M-PMV (**b**) cores was assessed as described in Figure 1. Plotted is the difference in infectivity between virus produced in the absence versus the presence of SERINC5 from at least three independent transfections. Statistical significance was determined via one-way ANOVA with Dunnett’s multiple comparisons post-test as for Figure 1. *, *p* < 0.05; **, *p* < 0.01; ****, *p* < 0.0001.

**Table 1 viruses-13-01279-t001:** Magnitude of SERINC5-mediated restriction and absolute infectivity for pseudotypes with the indicated viral glycoproteins on the indicated virion cores.

	HIV-1 Core				MLV Core				M-PMV Core			
Glycoprotein	Fold Restriction *^a^*	SEM	n *^b^*	Avg. Inf. (TU/mL) *^c^*	Fold Restriction *^a^*	SEM	n *^b^*	Avg. Inf. (TU/mL) *^c^*	Fold Restriction *^a^*	SEM	n *^b^*	Avg. Inf. (TU/mL) *^c^*
HIV-1	**132.9 ******	35.6	3	1.50 × 10^5^	**23.5 ****	10.6	3	7.50 × 10^3^	**360.1 ******	181.1	3	2.63 × 10^4^
ALV-A	3.6	1.0	4	9.30 × 10^6^	3.4	0.6	4	1.61 × 10^5^	3.9	1.2	4	4.32 × 10^5^
HERV-K	1.2	0.3	3	4.96 × 10^2^	3.2	1.4	5	2.43 × 10^2^	1.2	0.2	4	2.10 × 10^3^
RD114	**4.6 ****	0.7	4	1.19 × 10^7^	7.4	2.5	5	1.04 × 10^6^	**24.8 ******	10.9	6	9.40 × 10^5^
M-PMV	1.5	0.2	3	6.50 × 10^4^	**46.3 ******	18.7	6	1.76 × 10^5^	**104.2 ******	31.9	8	3.72 × 10^5^
EcoMLV	1.7	0.6	3	1.30 × 10^7^	2.3	0.6	4	6.19 × 10^5^	3.2	1.6	5	1.17 × 10^6^
A-MLV	**123.8 ******	22.7	12	1.68 × 10^6^	**61.7 ******	33.7	7	4.68 × 10^5^	**315.8 ******	87.9	12	9.80 × 10^5^
HTLV-1	1.2	0.4	3	1.66 × 10^3^	2.1	0.8	5	8.06 × 10^2^	1.1	0.2	4	2.60 × 10^3^
Flu (H7)	**27.3 ******	10.6	5	6.61 × 10^6^	**67.1 ******	27.2	4	5.90 × 10^4^	**31.2 ******	9.6	8	1.01 × 10^6^
PIV5	2.1	0.6	6	4.06 × 10^5^	2.7	0.9	3	2.04 × 10^4^	**9.6 ****	3.5	7	2.21 × 10^5^
Measles	2.0	0.2	4	5.49 × 10^4^	1.5	1.0	3	9.98 × 10^1^	0.9	0.1	4	9.01 × 10^3^
RABV	**12.0 ******	3.6	3	3.28 × 10^6^	**24.2 ***	10.8	6	2.46 × 10^4^	1.1	0.5	4	3.91 × 10^3^
LCMV	**4.1 ***	1.1	3	9.76 × 10^6^	1.2	0.1	3	1.19 × 10^5^	5.1	2.4	7	4.65 × 10^5^
MARV	1.9	0.5	3	9.08 × 10^6^	2.6	0.4	3	7.19 × 10^4^	3.1	0.8	4	2.67 × 10^5^
EBOV	1.9	0.1	3	3.64 × 10^5^	4.6	1.7	3	2.41 × 10^3^	1.7	0.2	4	2.50 × 10^4^
SARS CoV	0.9	0.2	5	1.80 × 10^4^	3.5	1.0	4	1.42 × 10^3^	1.0	0.2	4	5.11 × 10^3^
VSV	3.1	0.6	11	7.86 × 10^7^	1.1	0.2	6	2.38 × 10^6^	1.0	0.1	11	3.68 × 10^6^

*^a^* Pseudotypes with statistically significant SERINC5 restriction activity are highlighted in bold: *, *p* < 0.05; **, *p* < 0.01; ****, *p* < 0.0001. *^b^* Number of independent viral stocks assessed. *^c^* TU = Transducing Units.

**Table 2 viruses-13-01279-t002:** Virion core dependency of SERINC5 antiviral activity.

	HIV-1 Core vs.		MLV Core vs.
Glycoprotein	MLV Core	M-PMV Core	M-PMV Core
M-PMV	*p* = 0.0012 *^a^*	*p* < 0.0001	N.S. *^b^*
PIV5	N.S.	*p* = 0.0355	N.S.
Rabies	N.S.	*p* = 0.0118	N.S.
RD114	N.S.	N.S.	N.S.
LCMV	N.S.	N.S.	N.S.

*^a^* For each glycoprotein, Brown-Forsythe and Welsh ANOVA tests with Dunnett’s T3 multiple comparisons post-test compared fold reduction in infectivity with SERINC5 for each virion core against every other virion core. The *p*-value is reported for statistically significant pairwise comparisons. *^b^* N.S. = not significant (*p* > 0.05).

## Data Availability

Data is contained within the article.

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
