# Peer review of "Influence of Different Glycoproteins and of the Virion Core on SERINC5 Antiviral Activity"

_viruses, 2021, doi:10.3390/v13071279_

Round 1

Reviewer 1 Report

The authors sought to address the breadth of SERINC5 antiviral activity and assess whether the route of entry impacts the sensitivity of viral glycoproteins to the antiviral effects of SERINC5. They investigated whether the co-expression of SERINC5 during viral production could inhibit a variety of glycoprotein pseudotypes of HIV, MLV, or M-PMV cores. Using this system, The researchers tested the sensitivity of a number of retroviral Env glycoproteins as well as representative glycoproteins from the Arenavirus, Coronavirus, Filovirus, Rhabdovirus, Paramyxovirus, and Orthomyxovirus families and observed that the glycoprotein is a major determinant of SERINC5 sensitivity. These findings reveal that the interplay between virion core and glycoprotein is a determinant of virion sensitivity to SERINC5 antiviral activity.

Some questions:

1, there is only data with SERINC5 in Figure1, figure 3 and table 3. Add all data with or without SERINC5 in the results. (Or supplementary figures)

2, lines 253-267 missing.

3, did you try HIV-2 cores?

4, You only tried package process. Did you try to infect the cell lines with SERINC5? Maybe TZM-bl-SERINC5 cells.

Author Response

We wish to thank the reviewer for their efforts on behalf of our manuscript. Responses to their itemized comments are listed here:

  1. Fig 1 and 3 show the fold-restriction due to SERINC5. In other words, the infectivity without SERINC5, divided by the infectivity with SERINC5. Actual infectivity data in luciferase units for each glycoprotein is listed in Table 1.  Our manuscript does not have a Table 3.
  2. The missing text was apparently hidden under Table 1 in the Viruses Template. Hopefully we have now fixed the template so that the missing text is now visible.

  3. We did not try HIV-2 cores.
  4. While it certainly would be interesting to test the effect of SERINC5 on other phenotypes, it is the effects of SERINC5 on the assembly of infectious particles that has been best characterized in the literature, and therefore what we focused on here. In any case, TZM-BL-SERINC5 cells would not be restricted because these cells express a glycogag which inhibits SERINC5.

Reviewer 2 Report

In this paper Diehl et al examine an array of different envelope glycoproteins from a plethora of viruses with respect to their sensitivity to SERINC5-mediated restriction. They find that a number of envelope glycoproteins outside the retrovirus family is also sensitive to restriction. In addition, they show that the core of the virus oftentimes affects the susceptibility of the envelope glycoprotein to SERINC5-mediated restriction.  The novel aspects of the paper are 1) looking at the role of the core in SERINC5 restriction and 2) showing the expansive antiviral effect of SERINC5. However the pseudovirus system is artificial, it is possible that envelopes affected by SERINC5 may not be affected in the context of the actual native virions. Therefore, the findings shown here may not have significant physiological relevance.   

Comments

Page 1, line 18, not just Moloney gGag affects SERINC5 internalization, Friend gGag does the same. More correctly, it should be stated that “ecotropic MLV GlycoGag”

Page 2, line 64. In the case of EBOV Gp pseudoviruses, Ahi et al mBIO16, show in Figure 8, that SERINC5 is actually proviral and that gGag has no effect on the proviral effect of SERINC5. What do you think is the reason for the difference with your data in Figure 1b.

Page 2, lines 65-69 the authors mention that there may be a correlation between viral fusion and sensitivity to SERINC5. “Timilsina et al mBIO 2020” have shown that the mode of fusion (pH dependent vs pH-independent) is not important for SERINC5 restriction, it should be included here.

In page 5, lines 210-222. The authors are clearly showing that certain viral envelopes are more susceptible to SERINC5 restriction (e.g. IAV) vs other (e.g. MARV) and that the site of viral entry is not important for SERINC5 restriction. As they are using TZMBL cells, a HeLa-derived cell line, that is not the physiological cell target of many of the envelopes shown here, it would strengthen their argument if they did a control experiment with ammonium chloride showing that that the pH-dependent entry of the envelopes that require pH for entry (e.g. IAV) was affected in these cells.  

Lines 242-268 are missing.

In lines 304-345, the authors show that SERINC5 sensitivity of different envelopes is highly dependent on the core of the virus. Based on these findings then, it is possible that RABV envelope for example that is sensitive to SERINC5 restriction in the different cores, may not be sensitive in actual RABV virions, as the RABV virion core is very different from that of retroviruses. While this reviewer doesn’t necessarily think the authors should do extra experiments about it, they should at least address it.

In lines 393-399, when it comes to MPMV, this is a simian virus. Could it be that the human SERINC5 restricts it but the SERINC5 from the natural host of this virus might have no effect on this virus? I think it is a possibility that might explain the lack of a viral antagonist and should at least be discussed here.

In lines 435-449, the authors focus on the retroviral envelope glycoproteins (HIV, MLV and MPMV), but what about the nonretroviral envelopes that you identified. Are there different patterns in assembly of the viruses the SERINC5-sensitive envelope glycoproteins are derived from? Is there info on the lipid content of IAV for example, which is very potently affected by SERINC5? I think the reviewers need to mention

Author Response

We thank the reviewer for their thoughtful comments and their efforts on behalf of our manuscript.

1. Page 1, revised line 20, we have changed the text as suggested to say “ecotropic" MLV GlycoGag.

2. Page 2, revised line 65. The reviewer states that Ahi et al mBIO16, show in Figure 8, that SERINC5 increases infectivity and that gGag has no additional effect. Actually, the figure in question shows that, in the absence of SERINC5, gGag inhibits the infectivity of the Ebola GP pseudotype. In the presence of SERINC5, gGag is no longer inhibitory. Therefore, actually, the data in Ahi et al is in agreement with ours: in the absence of gGag, SERINC5 has no effect on Ebola pseudotypes.  

3. Page 2, revised lines 67-71: as suggested we now mention the results of “Timilsina et al mBIO 2020” and the reference is added to the end of the list of references on page 15, line 619.

4. page 5, revised lines 220-225. The reviewer suggested that we do a control experiment with ammonium chloride to show that that the pH-dependent entry of the envelopes that require pH for entry (e.g. IAV) was affected in these TZMbl cells. We have now done this experiment and include it as the new Figure 3.  

5. Lines 242-268 are missing. This text was underneath one of the Figures in the Viruses template and should now be visible.

6. As suggested by the reviewer, our statement on revised line 397 has been extended to say, "Further studies will be needed to determine whether the glycoproteins identified here as being restricted by SERINC5 in the context of pseudotypes – including IAV, RABV, PIV5, and LCMV - are inhibited by endogenous SERINC5 in the context of autologous virions."

7. Revised line 424: As suggested by the reviewer, we mention that if the simian virus M-PMV encodes a SERINC5 antagonist, it might not work in human cells.

8. Revised lines 476, the reviewer makes a great point here, that our comments regarding the possibility that lipid composition is a determinant of viral sensitivity to SERINC5 were focused only on retroviruses. We now mention the importance of lipid for IAV infectivity and pathogenicity as well. A new reference was added, and this is Reference #66 on page 15, line 645.

Reviewer 3 Report

The manuscript presented by Diehl and colleagues describes the influence of SERINC5 on the infectivity of several combinations between viral glycoproteins and virion cores.

The manuscript is well written and the experiments rather straightforward.

However, the study will benefit from a more in-depth analysis of some combinations.

The results of this study suggest that the processing status of some viral glycoproteins (both in terms of subunit processing and possibly cytoplasmic tail processing) may be affected according to the virion core to which they are combined with.

While this will be difficult to analyse for all the combinations presented here, this could be examined at least for a few of them (HIV-1 Env, RD114 and EBOV for example), improving the quality of the present work.

Minor

  • The lack of correlation between the absolute infectivity of the different pseudotypes and SERINC5 susceptibility should be shown as a separate Figure.  

Author Response

We greatly appreciate the efforts of the reviewer on behalf of our manuscript. 

The reviewer requested that we compare the processing status of the glycoproteins on the different virion cores. Unfortunately, this experiment is likely to be a full project in itself, beyond the scope of this manuscript, even with those few glycoproteins for which we have reagents.

The reviewer had a minor suggestion, which was to separate out panel C of the original Figure 2 (revised Figure 4) into a separate Figure. We prefer to keep this panel within this figure, especially since we put it there in response to feedback from several outside scientists. 

Reviewer 4 Report

Diehl at al have performed a very nice study showing that different viral env glycoproteins as well as different viral cores affect the antiviral activity of Serinc5. The experimental procedures are appropriate and the results are clearly described.

Major comments:

  1. In figure 1 the authors show the inhibition of infectivity by Serinc5 of virus pseudotyped with different envelopes. Serinc5 expression is regulated by a strong CMV promotor to minimize false negative results. Confirmation of high Serinc5 expression on the cell surface by flowcytometry should be included.
  2. Virus pseudotyped with M-PMV and PIV5 (figure 1a) and RESTV and TAFV (figure 1b) glycoproteins are reported to have reduced infectivity however these results are not significant. I would have concluded that Serinc5 did not significantly affect the infectivity of virus pseudotyped with these glycoproteins especially since Serinc5 is expressed at high levels during viral production. Is inhibition of infectivity of these pseudotypes significant when higher levels of Serinc5 are expressed during virus production?  
  3. The authors mentioned that nearly identical results were obtained using HIV env truncated after residue 710. These data are not shown and no explanation is given why this was included.
  4. In Figure 2 a positive controle (like HIV-1 Nef) is missing to show downregulation of Serinc5 from the cellular membrane. Moreover, the Serinc5 expression in the cell lysates seems very low in the case of HIV, HERV-K, PIV5, measles. Downregulation of Serinc5 from the cell surface should be shown by flowcytometry.
  5. In the section virus production and transductions: the protocols given should be more precise and phrases like “in the evening” and “the morning following..” should be avoided and replaced by time (hours) after transfection/transduction.
  6. The number format of cells seeded should be adjusted: 3x105 is most likely 300.000 cells.

Author Response

We appreciate the thoughtful comments of the reviewer and their efforts on behalf of our manuscript.

  1. There is no reason to check for SERINC5 expression on the cell surface by flow cytometry since we go beyond that by showing in Figure 2 that SERINC5 protein is incorporated in virions. It is the activity of SERINC5 within virions that is important for inhibition of virion infectivity, not cell-associated levels of SERINC5. Anyway, that being said, as we and others have already demonstrated high-level SERINC5 protein in the cells by multiple methods (Rosa et al Nature 526:212).
  2. As mentioned in point 1 above, we have already shown that the methods we are using here give high level expression of SERINC5 (Rosa et al Nature 526:212) and we do not believe it possible to get higher level SERINC5 without causing toxicity to the transfected cells.
  3. Rather than stating "data not shown", we have added a new figure which shows that identical inhibition by SERINC5 was observed with WT or cytoplasmic tail truncated HIV Env (new Figure 2, with new figure legend on line 275), and added text to explain the significance of this experiment (revised line 265) and a new reference (revised line 647)
  4. The experiments here are concerned with effects of virion associated SERINC5 on virion infectivity. If we added in Nef we would inhibit SERINC5 incorporation into virions. It is not clear to us what this would add to our story, or why Nef would be an appropriate negative control, since we have the condition with SERINC5 as a negative control.
  5. Revised line 111, 120, and 130: as suggested we have replaced vague statements like "overnight" with precise ones like 16 hrs.
  6. Revised lines 111 and 125 and The reviewer is correct that the exponents were not formatted correctly. This has been fixed.

Round 2

Reviewer 1 Report

Accept in present form

Reviewer 2 Report

A very interesting paper on SERINC5, examining a novel aspect of its biology. 

Reviewer 3 Report

In this revised form the authors have not addressed the single concern of this reviewer.

This reviewer does not believe that performing a WB on purified virions to detect the processing status of (a few well detected) Envelope glycoproteins represents a project in itself.

Reviewer 4 Report

The authors have adequately responded to the questions.